# Advances and Obstacles in Homology-Mediated Gene Editing of Hematopoietic Stem Cells

**DOI:** 10.3390/jcm10030513

**Published:** 2021-02-01

**Authors:** Christi T. Salisbury-Ruf, Andre Larochelle

**Affiliations:** Cellular and Molecular Therapeutics Branch, National Heart, Lung and Blood Institute (NHLBI), National Institutes of Health (NIH), Bethesda, MD 20892, USA; christi.gurniak@nih.gov

**Keywords:** gene editing, hematopoietic stem and progenitor cells, homology-directed repair, CRISPR-Cas9, AAV6

## Abstract

Homology-directed gene editing of hematopoietic stem and progenitor cells (HSPCs) is a promising strategy for the treatment of inherited blood disorders, obviating many of the limitations associated with viral vector-mediated gene therapies. The use of CRISPR/Cas9 or other programmable nucleases and improved methods of homology template delivery have enabled precise ex vivo gene editing. These transformative advances have also highlighted technical challenges to achieve high-efficiency gene editing in HSPCs for therapeutic applications. In this review, we discuss recent pre-clinical investigations utilizing homology-mediated gene editing in HSPCs and highlight various strategies to improve editing efficiency in these cells.

## 1. Introduction

The modification or insertion of genes was initially proposed in the early 1970s as a curative approach for inherited disorders [1]. Hematopoietic stem cells (HSCs) are preferred targets for genetic therapies owing to their ability to sustain lifelong hematopoiesis, affording the possibility of durably alleviating a range of conditions. Current gene therapy approaches for inherited blood disorders primarily entail the harvest of hematopoietic stem and progenitor cells (HSPCs) from individuals with an underlying genetic defect and their adoptive transfer after genetic modification ex vivo (Figure 1a). Decades of allogeneic HSPC transplantation performed in the clinic provided a roadmap for therapeutic translation of this novel approach. The avoidance of allo-reactive complications and the reduced complexity of conditioning regimens in autologous transplantation of genetically modified HSPCs provide substantial advantages over allogeneic HSPC transplantation options for these disorders.

Clinical trials using gene delivery vectors based on γ-retroviral vectors were initially approved in the 1990s, but only low numbers of corrected cells were detected, and phenotypic correction of the underlying defect was not observed. A refocus on optimizing ex vivo transduction conditions and addition of conditioning regimens to favor engraftment of transduced cells led to the first unequivocal clinical successes in patients with primary immunodeficiencies [2,3,4]. However, the subsequent reporting of malignancies caused by vector-mediated insertional activation of proto-oncogenes in treated patients [5,6,7] encouraged the development of alternative vector designs primarily based on retroviruses of the HIV-1 Lentivirinae subfamily (Figure 1b). Unique constituents of lentiviral vectors facilitate their nuclear translocation within non-dividing HSPCs, further enhancing transduction of these cells. The elimination of portions of the 3′-LTR promoter and enhancer elements in these vectors also provide a key self-inactivating (SIN) safety feature to alleviate concerns on possible recombination with endogenous HIV particles or unintended activation of proto-oncogenes near the genomic site of vector integration. For these SIN vectors, however, the efficiency of transgene expression is highly dependent on the addition of an internal ubiquitous promoter, and reduced or ectopic expression of the therapeutic gene can be limiting in disorders requiring robust or targeted transgene expression for a therapeutic effect.

In recent years, transformative advances have emerged to precisely edit cellular genomes (Figure 1c). Unlike integrating vectors, which can only facilitate gene addition within undefined loci of the cellular genome, novel editing strategies can mediate precise gene correction, gene ablation, and targeted gene addition within cells. Hence, these technologies further address safety concerns associated with integrating vectors, allow more robust and physiologic gene expression by targeted integration of transgenes near endogenous promoters, and extend gene therapies to dominant negative disorders requiring replacement of abnormal gene products rather than simple gene addition. Building on three decades of scientific advances and clinical successes using integrating viral vectors, targeted gene editing in HSPCs is now undergoing similar accelerated pre-clinical and clinical development. In this review, we summarize the current state of targeted gene delivery in HSPCs and examine new strategies developed to improve gene editing efficiency to levels necessary for effective treatment of inherited blood disorders.

## 2. The Process of Genome Editing

High-efficiency targeted genome editing in mammalian cells generally depends on the initial introduction of a DNA double-stranded break (DSB) at the chosen genomic locus to stimulate cellular DNA repair to yield desired outcomes. As summarized in this section, various cellular nucleases have been engineered to recognize individual target sequences and induce the necessary DSBs and DNA repair response for targeted DNA modification (Figure 1c). Alternative strategies to manipulate cellular genomes that do not rely on double-stranded DNA cleavage, including base editors [8,9,10,11], prime editors [12], and transposases/recombinases [13,14,15,16], were also developed in recent years and have been reviewed elsewhere [17].

### 2.1. Targeted DNA Double-Stranded Breaks with Engineered DNA Ucleases

Programmable DNA nucleases emerged in the late 1990s and early genome editing studies relied on protein guided synthetic zinc-finger nucleases (ZFNs) (Figure 1c) [18]. ZFNs consist of a non-specific FokI nuclease domain and a finger domain that provides DNA binding specificity. Each amino acid within the finger domain recognizes three DNA base pairs (bp), with several domains required to recognize a 9–18 bp motif. Specific DSBs are made upon dimerization of ZFNs at their FokI domains on opposite strands of the DNA. Zinc-finger nucleases were first shown to successfully edit drosophila DNA in 2002 [19] and subsequently in primary human T-cells in 2005 [20]. ZFNs have now entered clinical trials [21,22], but their widespread use has been hindered by constraints of the DNA-triplet recognition motif and the specialized expertise required to customize the DNA binding nuclease effector proteins for each genomic target site.

A more versatile transcription activator-like effector (TALE) DNA binding domain from the *Xanthomonas* spp. proteobacteria [23] was subsequently tethered to the same Fok1 endonuclease domain found in ZFNs to create TALE nucleases (TALENs) (Figure 1c) [24]. TALE domains are modular arrays of conserved repeats of 33–35 amino acids in length. Each repeat binds to a single nucleotide within the target sequence with a binding specificity dictated by the repeat-variable di-residue (RVD) at amino acid positions 12 and 13 of the TALE domain [25]. TALENs have been successfully used in pre-clinical models to edit HSPCs at the *CCR5* locus for treatment of HIV [26] and correct the sickle cell mutation in *HBB* with a single-stranded oligonucleotide (ssODN) donor template [27]. While TALENs’ RVD-DNA recognition code facilitates the design of binding domains with a broader targeting range than ZFNs, TALEN-based gene editing technologies still entail the complex assembly of nucleases specific to each targeted DNA locus.

The bacterial clustered regularly interspaced palindrome repeat (CRISPR) and the CRISPR-associated (Cas) protein, known as CRISPR/Cas, constitutes a novel class of RNA-guided programmable nucleases with unique simplicity and flexibility for targeted gene therapies (Figure 1c) [28]. Identified as a bacterial adaptive immune system [29], CRISPR destroys foreign DNA using the Cas endonuclease in a sequence-specific manner. These naturally occurring immune systems have been categorized as either CRISPR-Cas class 1, which requires complexes composed of several effector proteins for cleavage, or class 2, which allows cleavage of nucleic acids with a single effector domain. Due to their simpler requirements, systems based on class 2 have been favored for genome editing. Class 2 is further partitioned into types II (Cas 9), V (Cas 12), and VI (Cas 13). The type II CRISPR/Cas9 system derived from *Streptococcus pyogenes* (SpCas9) is currently the most widely used tool for genome editing in hematopoietic and other cellular sources. Cas9 is guided by a dual-RNA complex consisting of a universal trans-activating CRISPR RNA (tracrRNA) that recruits the Cas9 protein, and a CRISPR RNA (crRNA) with homology to a specific DNA sequence. The system was simplified for genome editing applications by synthetic fusion of both RNAs into a single guide RNA (gRNA). Small chemical groups may also be introduced at the extremities of synthesized gRNA to enhance gene editing, as shown at three therapeutically relevant loci in human HSPCs [30]. The Cas9/gRNA ribonucleoprotein (RNP) complex binds to a cognate proto-spacer adjacent motif (PAM) sequence (i.e., NGG) at the target locus, facilitating heteroduplex formation between the guide RNA sequence and the unwound target DNA strand. Cas9 then undergoes conformational changes, which activate its constituent HNH and RuvC nuclease domains to promote cleavage of both target (i.e., bound to the gRNA) and non-target DNA strands, respectively. The process results in formation of predominantly blunt-ended DSBs upstream of the PAM sequence at the chosen locus.

Several Cas9 variants or alternative Cas proteins have been developed to offset limitations of the CRISPR editing system based on SpCas9. For instance, off-target gene editing at unintended sites may result in deleterious cellular effects. Dual-strand targeting using paired Cas9 nickases derived by mutating the RuvC (Cas9^D10A^) or HNH (H840A) catalytic domains, and two adjacent gRNAs targeting opposing strands of a DNA target [28], can enhance CRIPR/Cas9 accuracy. Similarly, systems based on catalytically inactive Cas9 fused to Fok1 (fCas9), which require recruitment of two Fok1 domains for cleavage [31], can lower the probability of off-target editing. However, design of these systems is more complex, and efficiency is generally lower. Reduced off-target activity was also reported using Cas9 isolated from the alternative bacterial species *Streptococcus thermophilus* [32] and *Francisella novicida* (FnCas9) [33], and from type V CRISPR effector Cas12b derived from *Bacillus hisashii* (BhCas12b) [34]. In HSPCs, the high-fidelity (HiFi) Cas9 mutant improved the on-to-off target ratio when delivered as a purified protein [35], but the potential benefits of other engineered Cas9 variants remain to be determined, as they generally support lower on-target activity [27]. The large cargo size of the CRISPR/SpCas9 system represents another limitation of this technology, precluding packaging within some viral delivery vectors for gene therapy applications. More compact wild-type [36] and mutant [37] Cas9 nucleases from *Staphylococcus aureus* (SaCas9), Cas9 orthologs derived from *Campylobacter jejuni* (CjCas9) [38] and *Neisseria meningitidis* (NmCas9) [39], and type V Cas12e notable for its small size [40] were recently characterized to address this shortcoming. Another disadvantage of the CRISPR/SpCas9 system is the inherent NGG-PAM recognition requirement that limits Cas target site ranges. Several variants have been reported to expand the genome editing armamentarium, such as type V Cas12a nuclease that generally uses orthogonal T-rich PAM sequences [41], NmCas9 that recognizes pyrimidine-rich PAM sequences [39,42], a near PAM-less “SpRY” variant of the prototypical SpCas9 [43] and numerous other Cas effectors with altered PAM specificity [44,45].

### 2.2. Cellular Pathways for Repair of DNA Double-Stranded Breaks

In mammalian cells, DNA DSBs are repaired by classic non-homologous end joining (C-NHEJ), alternative NHEJ (alt-NHEJ, also known as microhomology-mediated end joining, MMEJ), single-strand annealing (SSA), and homology-directed repair (HDR) (Figure 2). The choice of DNA repair pathway after nuclease-mediated DSB formation is influenced by several factors that primarily coalesce on the key role that cell cycle plays in regulating DSB repair [46]. For instance, various phases of the cell cycle will differ in abundance or availability of pathway-specific DNA repair proteins and homologous DNA templates, and the repair mechanism favored may be influenced by the chromatin state of the target cells [47]. In genome editing applications, the DNA end structures induced by distinct programmable nucleases (i.e., blunt ends, 3′ overhangs, or 5′ overhangs) may also trigger distinct cellular pathways for repair of DSBs.

The classic form of NHEJ is operational throughout the cell cycle, except mitosis, and quiescent HSPCs largely rely on this mechanism to repair DSB lesions [48,49,50]. Unlike other DSB repair pathways that require DNA end resection at the break site to expose the homology required for repair, broken DNA ends containing no or limited sequence homology (0–4 bp) are ligated in C-NHEJ, and resection is thus not required. Both ends of DSBs are protected from extensive resection by high-affinity binding of Ku70/80 heterodimer complexes [51] and other end protection proteins including the DNA damage response TP53 binding protein 1 (53BP1) and its effectors (RAP1-interacting factor 1 [RIF1], CST complex-polymerase-α [CST-Polα] and the shieldin complex). Compatible DNA ends, such as blunt ends generated by Cas9, are often directly ligated by the XRCC4-DNA ligase 4 (XRCC4:LIG4), with the enhancing activity of XRCC4-like factor (XLF) or paralogue of XRCC4/XLF (PAXX). Incompatible ends, such as 5′/3′-overhangs or 3′-recessed DNA ends, require processing by the Artemis-DNA-PKcs nuclease complex to trim non-complementary end structures, or by Pol µ, Pol λ, and Tdt polymerases to add complementary nucleotides to favor XRCC4:LIG4-mediated ligation. In the absence of donor DNA, the original DNA sequence is generally restored, but limited sequence alterations (e.g., small indels) may also occur at the repair junctions, resulting in silent changes or frameshift mutations leading to target gene inactivation. When a donor sequence is added in vivo along with CRISPR/Cas and gRNA constituents, C-NHEJ can also mediate targeted integration at sites of Cas9-induced DSBs; however, a small percentage of stably integrated sequences may occur in the reverse (undesired) orientation (Figure 2).

The other DSB repair pathways (alt-NHEJ, SSA, and HDR) are known to be active during the S and G2 phases of the cell cycle. They share a first 5′-to-3′ end short-range resection step catalyzed by the Mre11/Rad50/Nbs1 (MRN) endonuclease complex in conjunction with the CtBP-interacting protein (CtIP). This step requires cyclin dependent kinases 1 and 2 (CDK1/2) to phosphorylate and activate CtIP, and is thus limited to the active phases of the cell cycle [52,53]. Recruitment of CtIP to MRN facilitates the removal of Ku70-Ku80 proteins from DSB ends and promotes the dephosphorylation of 53BP1, which in turn inhibits repair by C-NHEJ [54]. This process initially generates 3′ single-stranded overhangs. When short (2–20 bp, most often 3–8 bp) complimentary base pair microhomologies internal to both broken ends are exposed following resection, the broken ends can be repaired by the alt-NHEJ mechanism, involving the annealing of microhomologies, removal of extraneous heterologous DNA flaps by the XPF-ERCC1 endonuclease, fill-in synthesis of the flanking single-stranded regions by DNA Polθ, and sealing by DNA ligases I and III [55,56,57]. Because heterologous flaps flanking the annealed regions of microhomologies are cleaved and lost during alt-NHEJ repair, this pathway is inherently more mutagenic than the classic form of NHEJ. When a donor DNA is added, this repair mechanism can also be exploited for gene knock-in at targeted genomic loci (Figure 2).

In SSA- and HDR-mediated DSB repair, more extensive resection is required. These pathways are thus considerably slower than classical NHEJ or alt-NHEJ mechanisms. The Bloom syndrome protein (BLM)-DNA2 and exonuclease 1 (EXO1) mediate this process, and the replication protein A (RPA) binds the resultant single stranded DNA with high affinity to protect its integrity [58,59]. In SSA, the extended resection exposes longer (> 50 bp) sequences of homologies that are uniquely annealed in a RAD52-dependent manner. Similar to alt-NHEJ, the non-complementary tails are then removed by the XPF-ERCC1 endonuclease complex, and the remaining nicks are sealed by DNA ligase 1 [60]. In a normal cellular context, the SSA repair mechanism results in the obligate deletion of a larger sequence between homologous repeats and may promote chromosomal rearrangements (Figure 2) [60].

In HDR [61], RAD51 recombinase is recruited in an ATP-dependent manner to RPA-coated single-stranded DNA, forming a RAD51-DNA nucleoprotein filament. This process is mediated by BRACA2, which is recruited to DNA DSBs by PALB2 and BRAC1 in humans [62,63,64]. The RAD51 ssDNA filament then locates a homologous DNA template. The template is generally a double-stranded sister chromatid available in late S/G2 phases of the cell cycle [52,65,66], or can be provided exogenously in genome editing applications in the form of a double-stranded donor flanked by homology arms. A homology tract of more than 100 bp is typically required as a template to initiate repair by homologous recombination. When complementary ssODN are used as DNA donors, DSBs are processed by a distinct mechanism, the single-strand template repair (SSTR) pathway, which is independent of RAD51 but requires an operative Fanconi anemia (FA) pathway and at least two RAD51 paralogs (RAD51C and XRCC3) [67]. In HDR, the ssDNA filament then invades the homologous region to form a displacement (D)-loop where the template DNA is copied by DNA polymerase δ (Pol δ). The second DSB end is eventually captured by the invading strand, forming a DNA intermediate with two Holliday junctions. This structure undergoes gap repair DNA filling and ligation, and is ultimately resolved at both Holliday junctions in a non-crossover or crossover mode. In some cases, repair can occur by synthesis dependent strand annealing (SDSA), in which the newly replicated DNA dissociates from the template without the formation of a Holliday junction, or by break-induced replication (BIR), when the second DSB end is absent or cannot be found; however, the role of these pathways in genome editing has not been defined (Figure 2). Owing to the obligate use of a donor template sequence, HDR is considered error-free and is generally the preferred pathway for genome editing. Site- and orientation-specific integration at a chosen locus, either upstream of an endogenous promoter or within a safe harbor locus, is a commonly desired repair outcome for therapeutic applications. However, the low frequency of HDR in primary cells, especially long-term repopulating HSCs, remains a challenge to achieving high rates of targeted gene insertion by HDR [48,68].

### 2.3. Cellular Delivery of Gene Editing Tools

Safe and effective cellular delivery of engineered nucleases, gRNAs, and template sequences constitutes a key step in the process of gene editing. For ex vivo genome editing, approaches for the delivery of the required constituents within target cells can be broadly classified into viral vectors, electroporation, and cell-penetrating peptides [69]. In primary cells, including HSPCs, nucleases and the associated gRNAs are most effectively delivered by electroporation of mRNA molecules or as an RNP complex between gRNAs and the nuclease (e.g., Cas9) protein. Unlike transfection of DNA plasmid molecules, this approach results in limited cytotoxicity to HSPCs. In addition, the short half-life of the complex temporally limits the nuclease activity and the likelihood of genome editing at off-target loci [70].

Donor template delivery has also been a significant challenge for gene editing of HSPCs, as electroporation of dsDNA is highly toxic. Several alternative delivery platforms have been successfully been used, including ssODNs co-delivered with Cas9 [27,71,72,73]. For larger gene insertions, viral vectors are very effective, including integrase deficient lentivirus (IDLV) [74,75,76] and adeno associated virus serotype 6 (AAV6) [77]. There are several caveats, however, to the delivery of donor templates using virus-based systems, including DNA packaging capacity, which is limited by the viral capsid size, and off-target integration of viral genes [78].

## 3. Pre-Clinical Development of HDR-Mediated Gene Editing in HSPCS

Few genetic diseases, such as sickle cell disease (SCD), thalassemia, and Fanconi anemia (FA), can be cured by NHEJ-mediated incorporation of frameshift indels for disruption of open reading frames. However, the surgical precision provided by HDR-based gene correction or addition is preferred for most inherited disorders. In this section, we discuss preclinical studies utilizing HDR-mediated genome editing in HSPCs from healthy subjects or patients with inherited disorders amenable to treatment by transplantation of edited HSPCs (Table 1).

### 3.1. Sickle Cell Disease (SCD)

Targeting the causative sickle mutation within the *HBB* gene by eliciting HDR DNA repair mechanisms in HSPCs has been challenging. Several groups have shown highly successful rates (up to 50%) of editing in vitro in HSPCs derived from SCD subjects’ marrow or peripheral blood [27,74,75,79]. However, transplant studies in NOD/SCID/IL-2rγ^null^ (NSG) mice to assess in vivo engraftment and long-term function of edited HSPCs (derived from healthy subjects and edited to introduce the sickle mutation due to limited availability of HSPCs from SCD patients) have thus far achieved less than 5% editing in engrafted human cells in the absence of pre-transplant selection of edited HSPCs [27,74,79]. By applying a GFP reporter selection system, a median of 90% edited human cells was attained within the marrow of NSG mice after transplantation, but marker-free selection will be necessary for clinical translation of these therapies [79] (Table 1).

### 3.2. X-Linked Severe Combined Immunodeficiency (SCID-X1)

SCID-X1 is a primary immunodeficiency caused by mutations in interleukin-2 receptor common gamma chain (*IL2RG*) gene resulting in loss of T-cell, natural killer (NK) cell, and B-cell function. Although allogeneic bone marrow transplant is curative, HLA-donor matching and complications from graft-versus-host-disease present therapeutic challenges. Targeted gene addition within the *IL2RG* locus in CD34+ HSPCs, first shown by the Naldini lab, achieved efficiencies of approximately 6% in vitro using ZFNs and IDLV for template delivery [76]. A second study with an optimized ZFN design improved HDR-mediated editing in SCID-X1 patient HSPCs to over 20% in vitro. When the IDLV donor DNA vehicle was substituted with AAV6 donor vectors, up to 50% HDR-edited cells were observed within the CD34+ CD133− cellular compartment, with over 20% editing of HSPCs transplanted into NSG mice [80]. More recently, the Porteus lab used a Cas9 RNP/AAV6 editing system for targeted integration of a complete cDNA at the endogenous IL2RG translational start site [81]. In healthy donor cord blood (CB) or mobilized peripheral blood (mPB) CD34+ HSPCs, they achieved median in vitro HDR editing rates of 45% and significant persistence of edited cells long-term after transplantation in primary and secondary NSG animals. Importantly, marrow CD34+ cells from six independent SCID-X1 patients were also edited. Ex vivo editing frequencies were comparable to healthy donor HSPCs, and mice transplanted with cells from one patient had only corrected *IL2RG* cDNA within the spleen with significant multilineage correction compared to unedited HSPC [81] (Table 1). Globally, these studies provide proof-of-concept that current approaches for targeted integration at the IL2RG locus may enable correction of the SCID-X1 phenotype in affected patients.

### 3.3. X-Linked Chronic Granulomatous Disease (X-CGD)

The X-linked form of CGD (X-CGD) is a rare primary immunodeficiency caused by mutations in the *CYBB* gene that encodes gp91^phox^, a subunit of NADPH oxidase 2 (NOX2). Individuals born with CGD have inherited phagocyte dysfunction and increased susceptibility to bacterial and fungal microorganisms, formation of chronic granulomas, and poor wound healing. Initial attempts to edit X-CGD HSPCs used engineered ZFNs for targeted insertion of a functional gp91^phox^cDNA within the genomic “safe harbor” *AAVS1* locus [82]. Investigators achieved 15% gp91^phox^ protein expression in vitro and an average of 10.7% of human CD45 + (hCD45) cells in NSG bone marrow expressed gp91^phox^ eight weeks post engraftment [82]. A follow-up study used Cas9 and an ssODN donor DNA to correct a single base mutation (C676T) within the CYBB gene of CD34 + HSPCs from an X-CGD patient. In this approach, the repaired CYBB gene remains under the control of its endogenous promoter, avoiding concerns over suboptimal expression from an ectopic promoter. Approximately 31% gp91^phox^ expression was observed after myeloid differentiation of edited HSPCs in vitro. Transplant of *CYBB* C676T corrected HSPCs into NSG mice resulted in 16.5% of hCD45+ derived from edited CD34+ cells expressing gp91^phox^ within the mouse bone marrow with partial restoration of NOX2 activity [71] (Table 1). Overall, this study demonstrated feasibility of a targeted approach for gene mutation repair in a monogenic inherited disorder.

### 3.4. X-Linked Hyper-Immunoglobulin (Ig)M Syndrome (XHIM)

Adaptive immune cell function, particularly T- and B-cell interactions, relies in part on the association of CD40–CD40 ligand (CD40L). CD40L on T-cells, expressed after T-cell receptor (TCR) engagement, activates CD40 on B-cells, resulting in antibody class-switching and long-term memory response. Mutations in the *CD40LG* gene result in recurrent infections and low serum immunoglobulins in XHIM patients. Initial editing rates up to 46% within exon 1 of the endogenous *CD40LG* gene were achieved in activated CD4 + T-cells. Average editing up to 29% of XHIM patient CD4+ cells successfully restored CD40L expression post-T-cell activation, CD40 binding, and B-cell class switching [88]. While these findings potentially pave the way for an adoptive T-cell based therapy, editing of HSPCs from mPB of XHIM patients has also been achieved. Evaluation of *CD40LG* edited HPSCs in NSG mice 12–20 weeks post-transplant revealed an average integration of 4.4% across all editing platforms tested, comparable to editing rates achieved in other primary immune deficiencies (PIDs) [83] (Table 1).

### 3.5. Severe Congenital Neutropenia (SCN)

Severe congenital neutropenia syndromes are a group of PIDs impacting neutrophil function, caused by mutations inherited in an autosomal dominant, recessive, or X-linked manner. The most common causal gene, ELANE, encodes neutrophil elastase, and can harbor several hundred different mutations. Patients with SCN have severely decreased absolute neutrophil counts (ANC), resulting in frequent bacterial infections and a high risk of developing myelodysplastic syndrome. While treatment with G-CSF can improve neutrophil function and mobilization, gene editing has the potential to be curative. Tran and colleagues employed two different CRISPR/Cas9 RNPs in combination with AAV6 to repair both a specific ELANE^L172P^ point mutation as well as targeting exon 4, in which the majority of ELANE mutations are found [84]. HDR efficiencies of 30% were achieved at exon 4 of ELANE in HSPCs from both healthy and SCN subjects, and 20% HDR-mediated correction was observed at the mutant ELANE^L172P^ alleles in bone marrow CD34+ cells from affected patients. Neutrophils differentiated in vitro from edited HSPCs produced similar amounts of reactive oxygen species (ROS), exhibited normal phagocytosis, efficient bacterial killing, and production of neutrophil extracellular traps (NETs). At four weeks post-transplant of humanized NOG-EXL mice (expressing human GM-CSF and IL-3), only ELANE corrected HSPCs were able to functionally differentiate into neutrophils [84] (Table 1).

### 3.6. Wiskott–Aldrich Syndrome (WAS)

Wiskott–Aldrich Syndrome (WAS) is an X-linked primary inherited immunodeficiency characterized by microthrombocytopenia and lymphocyte dysfunction. Affected patients have increased incidence of infectious and autoimmune complications. Mutations in the *WAS* gene result in abnormal WAS protein (WASp), causing disruption of the actin cytoskeleton and impaired cellular mobility and interactions. Both CRISPR/Cas9 and ZFNs platforms were first shown to facilitate HDR-mediated gene editing at the WAS locus in a K562 cellular model [89]. In a subsequent study, Rai and colleagues used Cas9 RNP and AAV6 donor vectors to mediate targeted knock-in of a PGK-GFP expression cassette or a codon divergent promoterless WAS cDNA downstream of the ATG translational start codon of the WAS gene in primary HSPCs. In CD34+ cells from healthy subjects, more than half (58.8%) of HSCs sorted from the bulk cellular population expressed GFP at culture day 14, indicating stable HDR-mediated integration of the reporter cassette in the most primitive HSPC compartment. In mPB and BM CD34+ cells derived from individuals with WAS, targeted integration and expression of the WAS cDNA were detected in 46% of treated HSPCs, surpassing expression levels (33%) attained with the standard lentivirus-based gene transfer strategy previously used by these investigators. Transplant of edited WAS HSPCs into NSG mice revealed an average of 36% edited hCD45+ cells expressing WASp, with a substantial increase in percentages of B-, T-, and myeloid cells expressing WASp compared to HSPCs transduced with lentiviral vectors. Gene correction of up to 16% of targeted long-term repopulating-HSCs was also determined by secondary transplantation, providing comprehensive preclinical evidence of efficacy of a CRISPR-based gene editing approach for the treatment of subjects with WAS [85] (Table 1).

### 3.7. Lysosomal Storage Disorders

Inherited lysosomal storage disorders (LSDs) are a group of metabolic diseases in which enzyme deficiencies result in abnormal accumulation of metabolic biproducts. Mucopolysaccharidosis type I (MPSI) results from mutations in the iduronidase gene (*IDUA*). Affected patients develop severe muscle and neurological complications due to the build-up of glycosaminoglycan (GAG). A flexible Cas9/AAV6 platform based on HDR-mediated targeted integration of a PGK-IDUA cassette at the CCR5 safe harbor site recently showed ex vivo fraction of targeted alleles of 28% in healthy donor HSPCs that declined to 5–6% in NSG engrafted cells, indicating a marked decrease in engraftment potential after gene editing [86]. Nevertheless, edited IDUA HSPCs were able to phenotypically and functionally correct skeletal and neurological defects including a significant reduction of GAG excretion in NSG-IDUA^X/X^ mice, a novel mouse model of MPSI [86]. Using a similar Cas9/AAV6 approach, Pavani and colleagues targeted the lysosomal acid lipase (*LAL*) transgene, associated with the LSD Wolman disease, into the alpha-globin loci (*HBA1* and *HBA2*) of healthy donor HSPCs [87]. This approach achieved more than 50% knock-in efficiency, with 87% of burst-forming unit-erythroid (BFU-E) colonies showing *LAL* integration and secretion of LAL enzyme. Transplant into NSG mice resulted in engraftment of edited HSPCs, decreasing over 16 weeks to less than 10% of hCD45+ cells. Ex vivo erythroid differentiation of human CD34+ cells isolated from the BM of engrafted mice demonstrated LAL enzyme production in differentiated erythroblasts [87]. Together, these studies provide support to further develop HDR-based gene editing strategies in HSPCs for the treatment of lysosomal storage disorders (Table 1).

## 4. Strategies to Improve HDR-Mediated Gene Editing in HSPCs

### 4.1. Modulation of DNA Repair and Cell Cycle Pathways with Small Molecules

Addition of small molecules targeting DNA repair pathways or cell cycle regulators during the ex vivo editing process has been widely used to improve HDR-mediated gene editing in HSPCs. Small molecules have an advantage over other strategies due to their general ease of use and possible benefits regardless of the genetic target. The majority of small molecules that are currently being used fall within five general categories: small molecule inhibitors of NHEJ proteins, small molecules that directly promote HDR, molecules modulating the cell cycle, molecules targeting chromatin structure, and those with undefined mechanisms. As summarized in Table 2, small molecules have been used in a variety of mammalian cell lines with variable effects on HDR. Differences in species, reporter systems, which may or may not use a nuclease, and target genes are possible sources of variability. Importantly, very few of these small molecules have been tested in hematopoietic cells, and fewer in primary human CD34+ HSPCs in the context of CRISPR/Cas9. The following sections will focus primarily on small molecules previously tested in cells of hematopoietic origin.

Small molecule antagonists of the NHEJ pathway are commonly used to promote HDR, but benefits must be weighed against increased cellular toxicity and the potential loss of genomic integrity. DNA-PK inhibitors, including NU7026 and NU7441, can improve HDR rates in a number of different cell lines, such as human iPSCs and MEFs. Gene editing of the chronic myeloid leukemia (CML) cell line K562 and primary CD34+ HSPCs with an AAV2/6 donor and Cas9 nuclease in the presence of NU7441 improved the frequency of HDR approximately two-fold, but some cellular toxicity was observed [90]. NU7026 has also been tested, alone or in combination with other small molecules, in hematopoietic cells edited with Cas12a in the presence of an ssODN donor template [91]. This approach improved HDR editing in K562, CD4+ T cells, and CD34+ progenitors, but the total number of HDR-edited CD34+ cells did not exceed 0.63%, and viability was only 65%. More recently, the DNA-PK inhibitor M3814 was also shown to significantly improve editing after a Cas9-induced DSB in K562 cells without significant toxicity or unwanted chromosomal alterations [72]. To date, M3814 has not been tested in primary CD34+ HSPCs. The DNA ligase IV inhibitor, SCR-7, was also tested in K562 and primary CD34+ HSPCs edited with AAV6 and Cas9; unlike previous reports in human cancer cell lines and porcine fibroblasts, SCR-7 did not improve HDR editing in K562 cells or CD34+ HSPCs [27,90].

Small molecules that directly promote the HDR pathway have also been used for genome editing in HSPCs. RS-1 was first identified in a large scale screen of compounds shown to enhance the activity of Rad51 [92]. It was then shown to improve Cas9-mediated HDR in U2OS and HEK293 cells [93]. Cas9 gene editing of K562 cells with a CD45-GFP template in the presence of RS-1 improved HDR approximately two-fold, but this finding was not reproducible in primary CD34+ HSPCs [90]. Similarly, more recent studies in human iPSCs with inducible CRISPR/Cas9 [91] and K562 cells with Cas9 RNP targeting two different genes [73] showed no additional benefit of RS-1. Recently, 26 chemical compounds that share a core structure with RS-1 were tested for their ability to improve HDR in HEK293 cells. One compound, identified as chemical 26, improved integration of a puromycin resistance cassette within the *ATG5* gene locus 7.5-fold more than a DMSO control [94].

Cell cycle plays a critical role in DNA repair decisions after induction of DSB within the cellular genome. Quiescent HSPCs do not use HDR [95]. A two-day, low-density culture in medium supplemented with cytokines and small molecules that favor HSPC maintenance and self-renewal, such as prostaglandin E2 (PGE2) [96], StemRegenin1 (SR1) [97], and the pyrimidinone derivative UM171 [98] is thus routinely used to coax HSPCs into cycle and promote more efficient HDR [99,100]. However, HSPCs in the active phases of the cell cycle engraft poorly compared to quiescent cells [101]. A two-pronged strategy was recently proposed to address this quandary [95]. Cells were first allowed to cycle in culture medium containing cytokines to promote the accumulation of alleles corrected by HDR, and then reverted back to a G0 quiescent state to maintain stemness and long-term engraftment potential of edited cells. Quiescence was re-induced after gene editing by a three-day culture in medium supplemented with regulators of Wnt (CHIR9901) and mammalian target of rapamycin (mTOR) (rapamycin) pathways [102]. This approach increased the number of HDR editing events by five-fold up to almost 30% of alleles in HSPCs capable of long-term engraftment in immune-deficient recipient mice, suggesting that HDR-edited stem cells had re-entered G0 and sustained long-term hematopoiesis in vivo [95]. Several strategies have also focused on small molecules that directly modulate regulators of the cell cycle. Many commonly used compounds stall the cell cycle at G2/M-phase, including nocodazole [103], ABT-751 [104] and RO-3306 (RO) [68]. Only RO, which selectively inhibits CDK1, has been tested in peripheral blood stem cells. In combination with a G2/M-phase restricted Cas9-fusion enzyme, hGemCas9 (see Section 4.3), RO significantly decreased the percentage of NHEJ-mediated gene editing within *HBB*; however, no differences in HDR were observed [68]. Importantly, as with NU7441, RO treatment also significantly decreased cell viability. Likewise, it has been shown that addition of nocodazole to mouse LSK progenitors significantly increases apoptosis [105]. Recently, inhibition of the FA pathway repressor CDC7 by a small molecule, XL413, improved HDR in K562 cells regardless of the genetic target and with minimal cell toxicity [73]. Mechanistically, XL413 was shown to extend progression through S-phase of the cell cycle. Gene editing using an ssODN donor targeting the *HBB* locus in HSPCs also increased when cells were treated with XL413, and with minimal toxicity. Studies using alternative donor templates such as AAV6 will be necessary to determine if HDR can also be improved in HSPCs.

An emerging strategy to improve HDR gene editing entails targeting the structure of chromatin DNA with histone deacetylase (HDAC) inhibitors. Acetylation of histone lysine residues decreases their affinity to DNA, opening the chromatin for enhanced accessibility to repair proteins and transcription factors. Although not yet tested in HSPCs, the HDAC inhibitor valproic acid (VPA) was shown to improve Cas9-mediated HDR approximately two-fold in safe harbor *AAVS1* locus of human ESCs/iPSCs. Editing was further enhanced with overexpression of Rad51 in combination with VPA treatment [106]. Additional HDAC inhibitors including trichostatin A (TSA) and PCI-24,781 have also been shown to enhance CRISPR/Cas9-mediated gene insertion in porcine fetal fibroblasts; however, this effect was not specific to HDR, as NHEJ repair also increased [107].

Lastly, L755507, a β3-adrenergic receptor agonist, was identified in a screen to improve HDR-mediated gene insertion in mouse embryonic stem cells. Unlike the majority of small molecules that target DNA repair, the mechanism of L755507 remains unknown. HDR-mediated CRISPR editing of the *ACTA2* locus in K562 cells was slightly improved with L755507 [108]. This result was repeated in K562 using a Cas9 RNP with an AAV6 donor targeting the *CD45* locus [90]. As with other small molecules discussed above, efficacy in CD34+ HSPCs will ultimately determine potential clinical application.

### 4.2. Cas9 Fusion Proteins

Another strategy to increase HDR utilizes chemical conjugation of small proteins or effector domains to the Cas9 enzyme itself. This allows for direct delivery to the site of the double-stranded break, increasing their local concentration and potency. Conjugation of Cas9 to CtIP has been shown to improve HDR editing in HEK293T cells [109,110]. Importantly, while both studies show significantly increased HDR-mediated gene integration over Cas9 alone, HDR efficiency was highly dependent on the targeted genetic locus and gRNAs. Moreover, the oligomerization domain of CtIP was shown to have variable importance depending on the cell line [110]. These results emphasize the importance of testing this strategy in primary HSPCs.

As with small molecules, Cas9 fusion enzymes have also been used to inhibit the NHEJ pathway. Using a dominant negative mutant of 53BP1 called DN1S conjugated to Cas9, HDR-mediated gene insertion of CD18 cDNA improved from 26% to 51% at the *AAVS1* target locus in primary B-cells derived from patients with leukocyte adhesion deficiency type 1 (LAD-1). Over 20% of the positive cells were also shown to have bi-allelic gene integration. Importantly, NHEJ inhibition was local rather than global, an advantage of this strategy over small molecules [111].

In addition to targeting DNA repair pathways, Cas9-fusion with the protein Geminin has also been used to synchronize gene editing with the cell cycle. Geminin is ubiquitylated and degraded during late M-G1 phase by APC, and thus restricts Cas9 activity to the HDR permissive S/G2 phase [68,112]. Work by Donald Kohn’s group showed that using hGemCas9 enzyme to target the *HBB* in HSPCs significantly decreased NHEJ, however did not improve rates of HDR. Addition of the CDK1 inhibitor RO increased the HDR/NHEJ ratio four-fold in vitro by further reducing NHEJ; however, toxicity of RO will likely preclude its clinical use.

### 4.3. ssODN-Cas9 Conjugates

A large number of Cas9 conjugated to ssODN donor templates have recently been shown to be successful alternatives to viral delivery of template DNA. Similar to Cas9 fusion proteins, the benefit of this approach is delivery of the donor template directly to the site of the DSB, potentially increasing the frequency and the rate at which SSTR editing occurs. Multiple strategies for Cas9-fusion proteins exist, primarily differing in the design of the conjugation method. Examples include an RNA aptamer-streptavidin strategy called S1mplex [113], a Cas9-SNAP fusion protein that covalently links to a modified ssODN [114], a Cas9-HUH endonuclease fusion protein that associates with an unmodified ssODN [115], and a gRNA-donor DNA conjugate (gDonor) [116]. Recently Ling and colleagues developed a modified Cas9 in which a synthetic amino acid is conjugated to a short DNA adaptor to allow its association with an ssODN template. This DNA to DNA adaptor system, rather than protein to ssODN, allows for ease of conjugation. It is also an improvement over biotin-streptavidin methods, as the linker is not cleavable by proteases. Furthermore, non-chemically modified ssODNs can be used. The authors report a 10-fold increase in HDR in HEK293 cells and a three-fold increase in mouse zygotes [117]. While these strategies have yet to be tested in HSPCs for diseases such as sickle cell, where large gene insertions are not required, this may prove to be beneficial.

## 5. Strategies to Bypass HDR-Mediated Gene Editing

While significant progress has been made in recent years to increase frequencies of HDR alleles after gene editing, alternative strategies that rely on end-joining DNA repair pathways, namely NHEJ and alt-NHEJ, have also been proposed to bypass the inefficiencies of HDR-mediated approaches for precise gene editing.

### 5.1. Homology-Independent Targeted Insertion (HITI)

Homology-independent targeted insertion (HITI) uses an NHEJ-based homology-independent approach in which Cas9 mediates a DSB within both the targeted genomic locus and the complementary sequence of the donor DNA template. The linearized template can then be inserted at the genomic locus by NHEJ repair. Correct insertion of the template in the forward orientation destroys the Cas9 nuclease cut site, while reverse insertion retains the Cas9 sequence, which can be re-cut [118]. This method resulted in gene insertion in about 55% of transfected primary neurons, with greater than 90% of insertions resulting in precise editing. In vivo, AAV HITI template delivery in a rat model of retinitis pigmentosa, an inherited disorder causing blindness in humans, partially corrected both pathological disease and visual function [118]. More recently, in vivo HITI delivery using synthetic supramolecular nanoparticles (SMNP) successfully inserted the therapeutic gene Retinoschisis 1 (RS1) into mouse retinal tissue within the Rosa26 locus [119]. Our group showed that HITI can also be used for gene insertion in primary human HSPCs. Using a Cas9 RNP and AAV6 with a GFP-vector targeting exon 1 of the *ITGB2* locus, mutated in LAD-1, we could successfully achieve an average of 11% GFP positivity in HSPCs, persisting in culture up to four weeks post editing. Colony forming unit assays reveal that HITI does not result in lineage skewing, and HITI-edited HSPCs transplanted into immunodeficient NSG mice comprised an average of 21% of engrafted human CD45+ cells within the bone marrow 18 weeks post-transplantation [120]. These promising results indicate that HITI has the potential to complement HDR strategies currently used for HSPC gene editing.

### 5.2. Precise Integration into Target Chromosome (PITCh)

An alternative approach to gene insertion termed precise integration into target chromosome (PITCh) relies on the alt-NHEJ DNA repair pathway. PITCh vectors have small microhomology sequences to the target locus, which flank the template gene. Similarly to HITI, a CRISPR/Cas9 DSBs is made in both the target gene of interest and exogenous template, which is then integrated at the target site [121]. While showing a clear advantage over HDR in vivo in mouse liver and neurons [122], a direct comparison of PITCh to HITI in primary neurons in vitro revealed a significant benefit of the HITI approach. These differences are likely due in part to in vitro versus in vivo delivery methods as well as gene targets.

### 5.3. Homology-Mediated End Joining (HMEJ)

A hybrid approach termed homology-mediated end joining (HMEJ) flanks a gene of interest with 800 bp HAs rather than microhomology sequences and, similar to the HITI approach, positions Cas9 target sites outside the homology arms. Thus, the HMEJ strategy can take advantage of both the HDR and end-joining pathways to insert a gene of interest into the genomic cut site [123]. To our knowledge, neither HMEJ nor PITCh have yet been utilized in HSPCs; a direct comparison of homology-independent and dependent methods in HSPCs would be of great interest.

## 6. Conclusions

A burst of scientific advances have led to improved gene editing modalities in recent years. Various challenges remain to fully realize targeted gene therapies of HSPC disorders, including further improving editing efficiencies to levels necessary for overall clinical benefit, understanding innate and adaptive immune responses to editing effector molecules and donor templates, limiting the risks of editing at off-target sites, and addressing the restricted availability of these therapies in underdeveloped nations.

Whereas this manuscript emphasizes the impact of modulating DNA repair and cell cycle pathways to increase percentages of edited HSPCs, alternative strategies have also evolved to circumvent the shortcomings of current protocols. Notably, the recent inception of base editors [8,9,10,11] and prime editing approaches [12] that do not depend on cellular DSB repair mechanisms, although limited to alterations of small sequences, may favor high-efficiency editing in quiescent HSPCs. Development of safe and effective ex vivo expansion platforms for genetically modified long-term repopulating HSPCs could also provide a clinically valuable strategy to increase cell doses and therapeutic efficacy. However, the development of clinically relevant methodologies for expansion of adult HSPCs remains a challenging goal in clinical hematology. A more refined understanding of the target cell’s underlying biology is needed for this approach to gain full therapeutic momentum. In contrast, in disorders such as Fanconi anemia, where rare genetically corrected HSPCs exhibit a powerful in vivo growth advantage after transplantation, ex vivo expansion may not be required, and low-efficiency editing may suffice for therapeutic relevance in these patients [124,125,126].

Ex vivo genome editing of human HSPCs has entered clinical trials [21], but continued innovations will be necessary to provide new or optimized schemata for diseases requiring high-efficiency precision editing. The prospect of durable clinical benefits in HSPC disorders, however, justifies continued support for this evolving class of medicines.

## Figures and Tables

**Figure 1 jcm-10-00513-f001:**
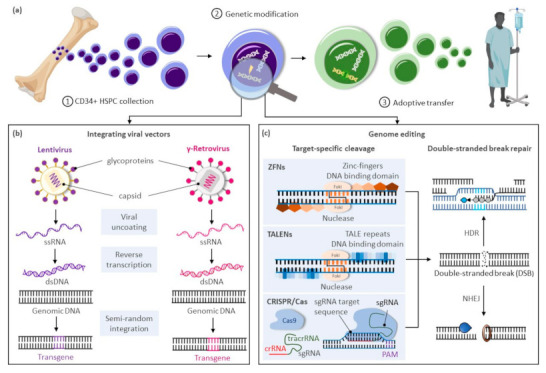
Gene therapy approaches for inherited blood disorders. (**a**) General scheme for gene therapies of inherited blood disorders: (**1**) Isolation of CD34+ hematopoietic stem and progenitor cells (HSPCs) from bone marrow harvests or mobilized peripheral blood cell collections; (**2**) Ex vivo genetic modification of HSPCs; (**3**) Adoptive transfer of genetically corrected cells to the patient generally following a reduced intensity conditioning regimen to enhance engraftment of the treated cells. (**b**) Largely random pattern of transgene integration within the target cellular genome after genetic modification of HSPCs using integrating viral vectors based on lentiviruses or gamma-retroviruses. (**c**) Precise integration of therapeutic genes using genome editing approaches based on zinc-finger nucleases (ZFNs), transcription activator-like effector (TALE) nucleases (TALENS), or the clustered regularly interspaced palindrome repeat (CRISPR)-associated (Cas) platform. Abbreviations: crRNA, CRISPR RNA; dsDNA, double-stranded DNA; DSBs, double-stranded breaks; HDR, homology directed repair; NHEJ, non-homologous end-joining; PAM, protospacer-adjacent motif; sgRNA, single guide RNA; ssRNA, single-stranded RNA; tracrRNA, trans-activating CRISPR RNA.

**Figure 2 jcm-10-00513-f002:**
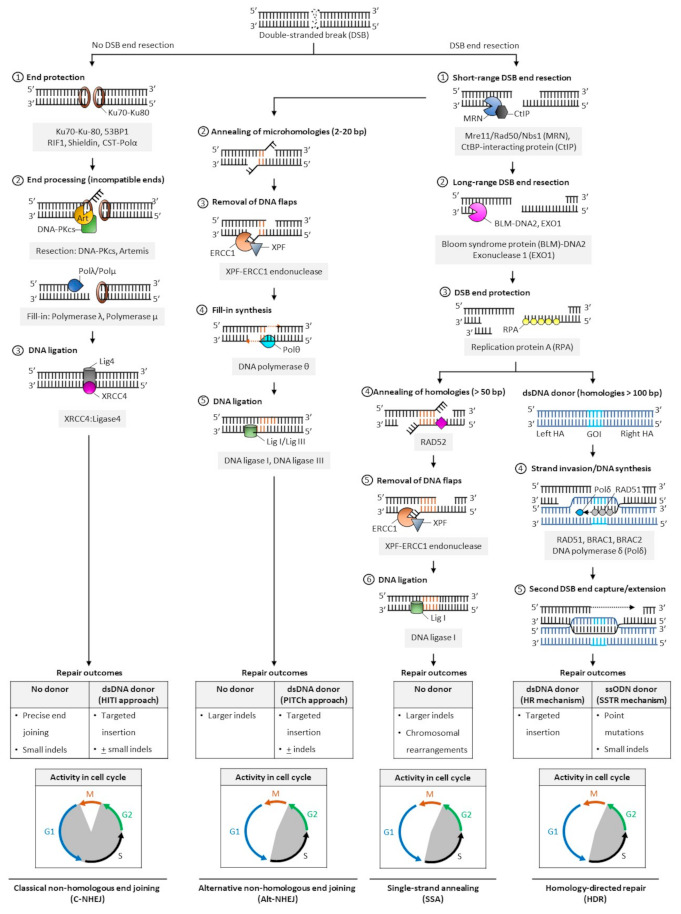
Pathways of DNA double-stranded break repair. Double-stranded breaks (DSBs) are introduced by engineered nucleases at the chosen genomic locus to stimulate endogenous cellular DNA repair mechanisms and promote various repair outcomes. In mammalian cells, DNA DSBs are repaired by classical non-homologous end-joining (C-NHEJ), alternative non-homologous end-joining (alt-NHEJ, also known as MMEJ), single strand annealing (SSA), or homology-directed repair (HDR) pathways. In the absence of donor templates, precise end joining, insertions and deletions (indels), and chromosomal rearrangements may be observed. Addition of a donor DNA template during repair can be used to install and correct point mutations, or knock-in larger DNA sequences. Classic NHEJ does not require template DNA and is the primary repair pathway in cells, whereas alt-NHEJ, SSA, and HDR are known to be active during the S and G2 phases of the cell cycle. Abbreviations: 53BP1: p53-binding protein 1; bp: base pairs; BRAC1: breast cancer type 1; BRAC2: breast cancer type 2; CtBP: C-terminal binding protein; DNA-PKcs: DNA-dependent protein kinase, catalytic subunit; dsDNA: double-stranded DNA; ERCC1: excision repair cross-complementation group 1; GOI: gene of interest; Mre11: meiotic recombination 11; Nbs1: Nijmegen breakage syndrome 1; RAD50/51/52: radiation sensitive 50/51/52; RIF1: Ras relate protein (Rap)-interacting factor 1; CST-Polα: CTC1-STN1-TEN1 (CST)-Polymerase α; HITI: homology-independent targeted insertion; HR: homologous recombination; indels: insertion and deletion; PITCh: precise integration into target chromosome; ssODN: single-strand oligodeoxynucleotide; SSTR: single strand template repair; XPF: xeroderma pigmentosum complementation group F; XRCC4: X-ray repair cross completing protein 4.

**Table 1 jcm-10-00513-t001:** HDR-mediated genome editing in HSPCs for inherited disorders.

Disease	Study	HSPC Source	Gene Editing System	Editing Efficiency at Target Locus	Ref.
SCD	In vitro	BM, SCD subjects	ZFN mRNA + IDLV	18.4% *HBB*	[74]
		BM, SCD subjects	Cas9 mRNA + IDLV	20% *HBB*	[75]
		Blood, SCD subjects	Cas9 RNP + ssODN	25% *HBB*	[27]
		mPB, SCD subjects	Cas9 RNP + AAV6	50% *HBB*	[79]
	In vivo	CB, healthy subjects	ZFN mRNA + IDLV	0.21% *HBB* (1° NSG BM) 0.27% *HBB (*1° NSG spleen)	[74]
		mPB, healthy subjects	Cas9 RNP + ssODN	2.3% *HBB* (1° NSG BM)	[27]
		mPB, healthy subjects	Cas9 RNP + AAV6	90% *HBB* (1° NSG BM + selection) 3.5% *HBB* (1° NSG BM-selection)	[79]
SCID-X1	In vitro	CB, SCID-X1 subject	ZFN mRNA + IDLV	>20% IL2RG	[80]
			ZFN mRNA + AAV6	>50% IL2RG	
		mPB/CB, healthy subjects	Cas9 RNP + AAV6	45% *IL2RG*	[81]
		mPB, SCID-X1 subjects	Cas9 RNP + AAV6	44.5% *IL2RG*	
	In vivo	CB, SCID-X1 subject	ZFN mRNA + IDLV	5% IL2RG (1° NSG BM HSPCs)	[80]
			ZFN mRNA + AAV6	>25% IL2RG (1° NSG BM HSPCs)	
		CB, healthy subjects	Cas9 RNP + AAV6	25.5% *IL2RG* (1° NSG BM) 9.5% to 20% *IL2RG* (2° NSG BM)	[81]
		mPB, SCID-X1 subjects	Cas9 RNP + AAV6	Corrected *IL2RG* (1° NSG spleen)	
X-CGD	In vitro	mPB, X-CGD subjects	ZFN mRNA + AAV6	15% AAVS1, gp91^ph^°^x^ expression in HSPC-derived myeloid cells	[82]
		mPB, X-CGD subjects	Cas9 mRNA + ssODN	31% AAVS1, gp91^ph^°^x^ expression in HSPC-derived myeloid cells	[71]
	In vivo	mPB, X-CGD subjects	ZFN mRNA + AAV6	10.7% AAVS1, gp91^ph^°^x^ (1° NSG BM)	[82]
		mPB, X-CGD subjects	Cas9 mRNA + ssODN	15.6% AAVS1, gp91^ph^°^x^ (1° NSG PB) 16.5% AAVS1, gp91^ph^°^x^ (1° NSG BM)	[71]
XHIM	In vitro	mPB, XHIM subjects	TALEN + AAV6	13.2% *CD40LG*, 5′ UTR	[83]
			Cas9 mRNA + AAV6	16.2% *CD40LG*, 5′ UTR	
			Cas9 RNP + AAV6	20.8% *CD40LG*, 5′ UTR	
	In vivo	mPB, XHIM subjects	TALEN + AAV6 ± adeno helper protein	Average of all editing strategies: 4.4% *CD40LG* (1° NSG BM). No increased editing with addition of adeno helper protein	[83]
			Cas9 RNP + AAV6 ± adeno helper protein		
SCN	In vitro	mPB, healthy subjects	Cas9 RNP + AAV6	30% *ELANE* (exon 4 gRNA)	[84]
		BM, SCN subjects	Cas9 RNP + AAV6	30% *ELANE* (exon 4 gRNA) 20% *ELANE* (L172P gRNA)	
	In vivo	BM, SCN subjects	Cas9 RNP + AAV6	3.1% *ELANE*-corrected neutrophils (1° NOG-EXL BM)	[84]
WAS	In vitro	mPB, healthy subjects	Cas9 RNP + AAV6	69% WAS, bulk CD34 + cells 67.3% *WAS*, sorted HSCs (day 7) 58.8% *WAS*, sorted HSCs (day 14)	[85]
		mPB/BM, WAS subjects	Cas9 RNP + AAV6	46.4% *WAS*, CD34 + cells (ddPCR) 45.5% WAS, CD34 + cells (flow)	
	In vivo	mPB/BM, WAS subjects	Cas9 RNP + AAV6	36.8% WAS (1° NSG BM) 16% *WAS* (2° NSG BM)	[85]
LSD	In vitro	CB/mPB, healthy subjects	Cas9 RNP + AAV6	28% *CCR5* (PGK-*IDUA*)	[86]
		mPB/CB, healthy subjects	Cas9 RNP + AAV6	~50% *HBA1* (*LAL* donor)	[87]
	In vivo	mPB/CB, healthy subjects	Cas9 RNP + AAV6	5–6% *CCR5, PGK-IDUA* (1° NSG BM)	[86]
		mPB/CB, healthy subjects	Cas9 RNP + AAV6	~8% *HBA1 (*1° NSG mice BM)	[87]

Abbreviations: AAV6: adeno-associated virus 6; BM: bone marrow; PB: peripheral blood; IDLV: integration-deficient lentivirus; tNGFR: truncated nerve growth-factor receptor; ssODN: single-stranded oligonucleotide; RNP: ribonucleoprotein; IL2R: interleukin-2 receptor common gamma chain; CB:, cord blood; mPB: mobilized peripheral blood; UTR: untranslated region; NSG: NOD/SCID/IL-2rγ^null^; NOG: NOD.Cg-Prkdcscid Il2rgtm1Sug/JicTac; NOG-EXL: NOD.Cg-Prkdcscid Il2rgtm1Sug Tg(SV40/HTLV-IL3, CSF2)10–7Jic/JicTac; HBB: beta-globin gene; CYBB: cytochrome B-245 beta chain; CD40LG: CD40 ligand; CCR5: C-C chemokine receptor type 5; IDUA: alpha-L-iduronidase; LAL: lysosomal acid lipase; SCD: sickle cell disease; ZFN: zinc-finger nuclease; TALEN: transcription-activator like effector nuclease; SCID-X1: X-linked severe combined immuno-deficiency; X-CGD: X-linked chronic granulomatous disorder; XHIM: X-linked hyper immunoglobulin (Ig)M syndrome; SCN: severe congenital neutropenia; WAS: Wiskott–Aldrich syndrome; LSD: lysosomal storage disorders; HSPCs: hematopoietic stem/progenitor cells; HBA1: hemoglobin A1; PGK: phosphoglycerate kinase; ELANE: elastase, neutrophil expressed.

**Table 2 jcm-10-00513-t002:** Small molecules tested in hematopoietic cells to improve HDR-mediated genome editing.

Molecule	Pathway	Molecule Target	Cells	Editing System	Target Locus	HDR Editing Results	Ref.
NU7441	NHEJ	DNA-PK inhibitor	K562	Cas9 RNP + AAV6	*CD45*	~2.5-fold increase	[90]
			CD34+	Cas9 RNP + AAV6	*CD45*	~2-fold increase	
NU7026	NHEJ	DNA-PK inhibitor	K562	Cpf1 + ssODN-TNS	*HPRT*	~4-fold increase	[91]
			CD4+	Cpf1 + ssODN-TNS	*HPRT*	3-fold increase	
			CD34+	Cpf1 + ssODN-TNS	*HPRT*	1.7-fold increase	
			K562	Cas9 RNP + AAV6	*CD45*	No increase	[90]
M3814	NHEJ	DNA-PK inhibitor	K562	Cas9 + ssODN	*FRMD7*	~4-fold increase	[72]
SCR7	NHEJ	Inhibitor of ligase IV	K562	Cas9 RNP + AAV6	*CD45*	No increase	[90]
			CD34+	Cas9 RNP + AAV6	*CD45*	No increase	
			CD34+	Cas9 RNP + ssODN	*HBB*	No increase	[27]
RS-1	HDR	RAD51 agonist	K562	Cas9 RNP + AAV6	*CD45*	~2.1-fold increase	[90]
			CD34+	Cas9 RNP + AAV6	*CD45*	No increase	
			K562	Cas9 RNP + dsDNA	*LAMP1* *FBL*	No increase	[73]
Nocodazole	Cell cycle	Inhibition of microtubule polymerization	K562	Cas9 RNP + dsDNA	*LAMP1* *FBL* *RAB11A* *TOMM20*	~5–10% increase in FBL, RAB11A, and TOMM20	[73]
RO-3306 (RO)	Cell cycle	CDK1 inhibitor	CD34+	hGemCas9 + AAV6	*HBB*	Increase in ratio HDR/NHEJ: 4-fold (in vitro), 7-fold (in vivo, NSG mice)	[68]
XL413	Cell cycle	CDC7 inhibitor, extends S-phase	K562	Cas9 RNP + dsDNA	*SMC1A* *LAMP1 HIST1H2BJ* *NPM1* *FUS* *TOMM20* *FBL* *RAB11A*	~1.6 to 3.5-fold increase at all loci	[73]
			T-cells	Cas9 RNP + dsDNA	*RAB11A* *TUBA1B* *CLTA*	Increase at all loci in a dose dependent manner	
			CD34+	Cas9 RNP + ssDNA	*HBB* *TOMM20*	Slight increase	
L755507	Unknown	β3-adrenergic receptor agonist	K562	Cas9 RNP + AAV6	*CD45*	~1.5-fold increase	[90]

Abbreviations: mPB-HSPC: mobilized peripheral blood-HSPC; TNS: targeted nucleotide substitutions; RNP: ribonucleoprotein; ssODN: single stranded oligonucleotide; PBMCs: peripheral blood mononuclear cells; NSG: NOD/SCID/IL-2rγ^null^; CDC7: cell division cycle 7; CDK1: cyclin-dependent kinase 1; DNA-PK: DNA-protein kinase; HBB: beta-globin gene; HPRT: hypoxanthine phosphoribosyltransferase 1; FRMD7: FERM domain containing 7; SMC1A: structural maintenance of chromosomes 1A; LAMP1: lysosomal associated membrane protein 1; HIST1H2BJ: histone cluster 1 H2B family member J; NPM1: nucleophosmin 1; FUS: FUS RNA binding protein; TOMM20: translocase of outer mitochondrial membrane 20; FBL: fibrillarin; TUBA1B: tubulin alpha 1b; CTLA: cytotoxic T-lymphocyte associated protein 4.

## Data Availability

Data sharing not applicable.

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
