# Peer review of "Advances and Obstacles in Homology-Mediated Gene Editing of Hematopoietic Stem Cells"

_jcm, 2021, doi:10.3390/jcm10030513_

Round 1

Reviewer 1 Report

This is a comprehensive and well written review summarizing major advances in genomic editing of HSCs. Authors cover basic aspects of nuclease-mediated gene editing, different cellular pathways for DNA repair, delivery of gene editing tools, preclinical and clinical developments in gene editing and various strategies to improve gene editing in HSCs.

The review is accompanied by appropriate illustrations, logical and easy to read. Review provides a balanced account of past research and new perspectives.   Overall, this review will appeal to a broad audience of researchers who are exploring gene editing in HSCs and beyond.

I have very few minor comments.

  1. Alternative nonhomologous end-joining (Alt-NHEJ) is also known as microhomology-mediated end joining (MMEJ). Please provide alternative name for Alt-NHEJ on line 167.
  2. Porteus and others recommend adding HSC agonists UM171 and/or SR1 to medium for culture of HSPCs after gene editing procedure. It would be valuable to discuss whether this strategy is helpful to improve engraftment of gene-edited HSPCs.
  3. Text in Lines 172-180 is displaced figure legend.
  4. Please correct duplicated text in Lines 170-171 and 181-182.

Author Response

Comments-Reviewer 1:

“This is a comprehensive and well written review summarizing major advances in genomic editing of HSCs. Authors cover basic aspects of nuclease-mediated gene editing, different cellular pathways for DNA repair, delivery of gene editing tools, preclinical and clinical developments in gene editing and various strategies to improve gene editing in HSCs.

The review is accompanied by appropriate illustrations, logical and easy to read. Review provides a balanced account of past research and new perspectives. Overall, this review will appeal to a broad audience of researchers who are exploring gene editing in HSCs and beyond.

I have very few minor comments.”

We thank the reviewer for their careful reading of our review manuscript. We have addressed the minor concerns in the manuscript and in the comments below.

Reviewer 1, Comment-1:

“Alternative nonhomologous end-joining (Alt-NHEJ) is also known as microhomology-mediated end joining (MMEJ). Please provide alternative name for Alt-NHEJ on line 167”

The alternative nomenclature (i.e., MMEJ) is used at the beginning of section 2.2 (line 154). For further clarification, the statement “alt-NHEJ, also known as MMEJ” was added to Figure (2) legend caption (line 170).

Reviewer 1, Comment-2:

“Porteus and others recommend adding HSC agonists UM171 and/or SR1 to medium for culture of HSPCs after gene editing procedure. It would be valuable to discuss whether this strategy is helpful to improve engraftment of gene-edited HSPCs”

Text was added to emphasize the importance of culture/expansion with HSC agonists, such as UM171 and SR1, before (section 4.1, lines 592-597) and after (section 5-conclusions, lines 744-748) gene editing procedures in HSPCs. 

Reviewer 1, Comment-3:

“Text in Lines 172-180 is displaced figure legend”

Figure (2) legend caption was reformatted (lines 167-183).

Reviewer 1, Comment-4:

“Please correct duplicated text in Lines 170-171 and 181-182”

Figure (2) legend caption was edited to address this issue (lines 173, 174).

Reviewer 2 Report

The review by Larochelle describes preclinical efforts to develop and improve gene editing strategies for human HSPC. Many important topics are encompassed, including the basic cell biology of gene editing and clinical translation efforts. Overall, I find the manuscript well written and comprehensive, although too lengthy and with unnecessary technical details. The narrative can be extensively trimmed and polished to make it look less like a thesis chapter and more like a review.

Many details need to be revised to overall improve the quality of the manuscript:

  • Page 3: The authors are wrongly citing reference 20. Urnov and colleagues edited IL2RG in WT T cells (SCID-X1 patients completely lack T cells).
  • Page 3: Synthesized gRNA may also be conjugated to small chemical groups to enhance gene  editing. I would rather say chemical modifications are introduced at the extremities of sgRNAs (they are not chemically conjugated).
  • Page 4: Lower HiFi Cas9 activity is site-dependent. Moreover, novel HiFi Cas9 mutants which retain high on-target activity have recently been developed. https://doi.org/10.1038/s41591-018-0137-0
  • Page 4: references for PAM-engineered Cas9 variants need to be updated.
  • In my opinion Figure 2 would be improved by addition of ssODN/ dsDNA integration scheme.
  • Page 7. At the end of the chapter authors state that SSTR is highly effective in human HSPCs in contrast to HDR, but reference a work where efficiencies of editing range in 10-20% in vitro and much lower upon transplantation. Authors need to rephrase the sentence.
  • Page 8. The authors state some work suggests that IDLV is more toxic to  HSPCs than AAV6. There is a much larger literature instead describing how lentiviral vectors are much well tolerated by HSC, including plenty of data from ongoing clinical trials that show stable polyclonal reconstitution. Extensive p53-mediated responses have been measured in gene edited HSC in response to AAV treatment and shown to be detrimental for HSC repopulation. DOI: 10.1016/j.stem.2019.02.019
  • Table I can be substantially improved and shortened by subsetting it in several columns (e.g. Disease- HSC source- type of editors- donor template- in vitro % editing- in vivo % editing- are the current efficiencies compatible with potential clinical benefit). Many of the details present in the table,such as the % of editing in vitro and in vivo, are also repeated in the main text. It would be helpful for the readability of the paper to almost completely remove them from main text.
  • Page 10. The authors state that their strategy using ZFNs and IDLV for template delivery was particularly inefficient in primitive HSPCs. Efficiencies were low after in vivo transplant, but that was also the first demonstration of HDR-mediated editing in HSC. All the subsequent literature built on that work to improve the editing protocol efficiencies and this should be acknowledged by the authors.
  • Page 11. Authors should remove this misleading sentence: Cas9, ZFN, and template delivery however were achieved by nucleofection, a strategy which is not feasible in HSPCs .
  • Page 12. Authors should mention that whereas one strategy to improve yield of edited HSC is to modulate DNA repair and thus increase the % of edited cells, drugs that expand HSC in culture has proven equally if not more critical in improving the yield of the total edited population and thus number of edited clones engrafted in vivo.
  • I find very relevant to include in the work, once it has been shortened, a paragraph describing Base editing as an alternative to DSB-mediated editing and the recent efforts towards its use in HSPC.

Author Response

Comments-Reviewer 2:

“The review by Larochelle describes preclinical efforts to develop and improve gene editing strategies for human HSPC. Many important topics are encompassed, including the basic cell biology of gene editing and clinical translation efforts. Overall, I find the manuscript well written and comprehensive, although too lengthy and with unnecessary technical details. The narrative can be extensively trimmed and polished to make it look less like a thesis chapter and more like a review."

We have trimmed and streamlined the text significantly, namely in sections 3 and 4 of the manuscript.

"Many details need to be revised to overall improve the quality of the manuscript”

We thank the reviewer for their careful reading of our review manuscript. We have addressed all concerns in the manuscript and in the comments below.

Reviewer 2, Comment-1:

“Page 3: The authors are wrongly citing reference 20. Urnov and colleagues edited IL2RG in WT T cells (SCID-X1 patients completely lack T cells).”

We have removed the incorrect statement: “from X1 SCID patient” (line 90). 

Reviewer 2, Comment-2:

“Page 3: Synthesized gRNA may also be conjugated to small chemical groups to enhance gene  editing. I would rather say chemical modifications are introduced at the extremities of sgRNAs (they are not chemically conjugated).”

We have replaced the statement “chemical conjugation” with the text suggested by reviewer (lines 119, 120).

Reviewer 2, Comment-3:

“Page 4: Lower HiFi Cas9 activity is site-dependent. Moreover, novel HiFi Cas9 mutants which retain high on-target activity have recently been developed. https://doi.org/10.1038/s41591-018-0137-0”

The novel HiFi Cas9 with high on-target activity is discussed in section 2.1 (lines 138, 139 and reference 35).

Reviewer 2, Comment-4:

“Page 4: references for PAM-engineered Cas9 variants need to be updated.”

We removed reference “Anzalone et al. Nat Biotech 2020” and added the following references: 1) Cas12a variant: Kleinstiver et al. Nature 2015 (line 149); 2) Other Cas effectors with altered PAM specificity: Chatterjee et al. Nat. Biotech 2020; Legut et al. Cell Rep 2020 (line 151).

Reviewer 2, Comment-5:

“In my opinion Figure 2 would be improved by addition of ssODN/dsDNA integration scheme”

We attempted addition of the ssODN/dsDNA integration scheme to Figure 2 but the resultant figure becomes too complex and illegible. We suggest retaining Figure 2 in its current form. Information on the ssODN/dsDNA integration scheme is clearly outlined in the text (section 2.1)

Reviewer 2, Comment-6:

“Page 7. At the end of the chapter authors state that SSTR is highly effective in human HSPCs in contrast to HDR, but reference a work where efficiencies of editing range in 10-20% in vitro and much lower upon transplantation. Authors need to rephrase the sentence.”

The statement and associated reference were removed (lines 253, 254).

Reviewer 2, Comment-7:

“Page 8. The authors state some work suggests that IDLV is more toxic to  HSPCs than AAV6. There is a much larger literature instead describing how lentiviral vectors are much well tolerated by HSC, including plenty of data from ongoing clinical trials that show stable polyclonal reconstitution. Extensive p53-mediated responses have been measured in gene edited HSC in response to AAV treatment and shown to be detrimental for HSC repopulation. DOI: 10.1016/j.stem.2019.02.019”

The statement and associated reference were removed (lines 269-271).

Reviewer 2, Comment-7:

“Table I can be substantially improved and shortened by subsetting it in several columns (e.g. Disease- HSC source- type of editors- donor template- in vitro % editing- in vivo % editing- are the current efficiencies compatible with potential clinical benefit). Many of the details present in the table, such as the % of editing in vitro and in vivo, are also repeated in the main text. It would be helpful for the readability of the paper to almost completely remove them from main text.”

Table 1 was substantially modified per recommendations from the reviewer (see revised Table 1 starting on line 298). The corresponding text in section 3 was also considerably revised for concision and to avoid repetitive details (see section 3 starting on line 274).

Reviewer 2, Comment-8:

“Page 10. The authors state that their strategy using ZFNs and IDLV for template delivery was particularly inefficient in primitive HSPCs. Efficiencies were low after in vivo transplant, but that was also the first demonstration of HDR-mediated editing in HSC. All the subsequent literature built on that work to improve the editing protocol efficiencies and this should be acknowledged by the authors.”

The statement on inefficiencies of initial studies using ZFNs and IDLV was removed. Text was clarified to indicate that lower editing efficiencies after in vivo transplantation have been observed with all editing platforms in preclinical studies for sickle cell disease (see section 3.1 starting on line 311).

Reviewer 2, Comment-9:

“Page 11. Authors should remove this misleading sentence: Cas9, ZFN, and template delivery however were achieved by nucleofection, a strategy which is not feasible in HSPCs.”

The misleading statement was removed (lines 493, 494).

Reviewer 2, Comment-10:

“Page 12. Authors should mention that whereas one strategy to improve yield of edited HSC is to modulate DNA repair and thus increase the % of edited cells, drugs that expand HSC in culture has proven equally if not more critical in improving the yield of the total edited population and thus number of edited clones engrafted in vivo.”

Text was added to emphasize the importance of culture/expansion with HSC agonists, such as UM171 and SR1, before (section 4.1, lines 592-597) and after (section 5-conclusions, lines 744-748) gene editing procedures in HSPCs. 

Reviewer 2, Comment-11:

“I find very relevant to include in the work, once it has been shortened, a paragraph describing Base editing as an alternative to DSB-mediated editing and the recent efforts towards its use in HSPC.”

A statement on alternative strategies to manipulate cellular genomes (such as base editing) can be found in the introductory paragraph of section 2 (lines 79-82). The importance of these alternative approaches was also reiterated in the final section of this review (section 5, conclusions, lines 739-743).